# Molecular Breeding for Fungal Resistance in Common Bean

**DOI:** 10.3390/ijms262110387

**Published:** 2025-10-25

**Authors:** Luciana Lasry Benchimol-Reis, César Júnior Bueno, Ricardo Harakava, Alisson Fernando Chiorato, Sérgio Augusto Morais Carbonell

**Affiliations:** 1Centro de Pesquisa em Recursos Genéticos Vegetais, Instituto Agronômico, Campinas 13075-630, SP, Brazil; 2Divisão Avançada de Pesquisa e Desenvolvimento em Sanidade Agropecuária do Instituto Biológico, Campinas 13101-680, SP, Brazil; cesar.bueno@sp.gov.br; 3Instituto Biológico, São Paulo 04014-900, SP, Brazil; ricardo.harakava@sp.gov.br; 4Centro de Grãos e Fibras, Instituto Agronômico, Campinas 13075-630, SP, Brazil; alisson.chiorato@sp.gov.br (A.F.C.); sergio.carbonell@sp.gov.br (S.A.M.C.)

**Keywords:** plant pathogen interaction, fungal disease, genetic resistance, quantitative resistance *loci* (QRL) mapping, plant immunity

## Abstract

Despite the recognized social and economic importance of common beans (*Phaseolus vulgaris* L.), the average grain yield is far below the productive potential of cultivars. This situation is explained by several factors, such as the large number of diseases and pests that affect the crop, some of which cause significant damage. It is estimated that approximately 200 diseases can significantly affect common beans. These can be bacterial, viral, fungal, and nematode-induced. The main bean fungal diseases include anthracnose, angular leaf spot, powdery mildew, gray mold, *Fusarium* wilt, dry root rot, *Pythium* root rot, southern blight, white mold, charcoal rot and rust. This review provides a comprehensive overview of eleven major fungal diseases affecting common bean, describing their associated damage, characteristic symptomatology, and the epidemiological factors that favor disease development. It further synthesizes current knowledge on host resistance mechanisms that can be exploited to develop molecularly informed resistant genotypes. The compilation includes characterized resistance genes and mapped quantitative trait *loci* (QTLs), with details on their chromosomal locations, genetic effects, and potential for use in breeding. Moreover, the review highlights successful applications of molecular breeding approaches targeting fungal resistance. Finally, it discusses conclusions and future perspectives for integrating advanced genetic improvement strategies—such as marker-assisted selection, genomic selection, gene editing, and pyramiding—to enhance durable resistance to fungal pathogens in common bean. This work serves as both a reference for forthcoming resistance-mapping studies and a guide for the strategic selection of resistance *loci* in breeding programs aimed at developing cultivars with stable and long-lasting fungal resistance.

## 1. Introduction

Among the commercial species of the Fabaceae family, the common beans (*Phaseolus vulgaris*; 2n = 2x = 22) stand out as the most important grain used directly for human consumption [1]. In developing countries, mainly Latin America and eastern Africa, beans are the basis of the diet of more than 300 million people, being the main source of protein and the second largest energy source [2].

The relevance of the bean for the food security of the planet is even greater given the trend of increased world population, especially in developing countries, which are not only the largest consumers but also the main bean producers [1].

The evolutionary process and domestication in common beans are among the most peculiar and studied among cultivated species. Originally from Mexico [3], the species expanded to South America, resulting in the formation of two gene pools called Mesoamerican and Andean [4]. Domestication occurred only after this subdivision, in at least one independent event within each pool, making the species a unique model for plant domestication studies [3].

The incidence of diseases in common beans severely affects the productivity of the crop [5]. Infectious diseases are responsible for losses that can reach levels of up to 100% of the cultivar. The application of conventional agrochemicals remains the main approach to control bean diseases. However, every method of control has its positive and negative aspects.

Regarding the financial impact of fungal diseases on common beans, Ramasubramaniam et al. [6] evaluated the economic losses in the field caused by white mold (*Sclerotinia sclerotiorum*) in North Dakota (USA) between 2003 and 2005. In the study, they estimated an average loss of 0.8% of the potential yield of dryland beans. Over the three-year period, they calculated an average annual economic loss of approximately US$ 2.95 million.

In 2024, Brazil produced 3,018,459 tons of common bean grains, valued at approximately US$ 65 million, from a harvested area of 2,631,805 hectares [7]. Losses caused by fungal diseases can have a significant impact, as shown by the study of Ramasubramaniam et al. [6], particularly when control measures are ineffective or costly or when disease pressure in the field is high, affecting both production costs and the market value of the grain.

In vegetative stages, the bean has a greater predisposition to the attack of death-causing fungi. Aerial pathogens occur more often during the period of flowering and fruiting, steps that require a high energy cost, which, at the same time, expend more energy to react to infections by activating their (structural and biochemical) defense mechanisms, in those susceptible cultivars [8].

Plants in initial vegetative stages release exudate that directs the attack of soilborne phytopathogenic fungi, which is the highest predisposition due to less lignification of the tissues. Infections result in the death of seeds in germination and overturning and death of the seedlings, common among the genera *Fusarium*, *Rhizoctonia*, *Sclerotium*, *Macrophomina* and *Pythium*. Among these, the most important in terms of damage to the crop are *Fusarium oxysporum* [8] and *Sclerotinia sclerotiorum* [9]. Aerial phytopathogens are more dependent on climatic precipitation factors, relative humidity and temperature. Severe yield losses in common bean are primarily attributed to infections caused by *Colletotrichum lindemuthianum* and *Pseudocercospora griseola* [10].

The cultivation of resistant cultivars developed through systematic plant breeding programs constitutes an effective and sustainable strategy for the management of diseases caused by *C. lindemuthianum*, *P. griseola*, and *F. oxysporum*. This approach neither increases production costs nor disrupts the routine practices of farmers. Moreover, it eliminates the need for fungicide applications, thereby reducing overall production expenses, mitigating environmental impacts, and minimizing the development of phytopathogen populations resistant to chemical active ingredients. Varietal resistance has demonstrated consistent effectiveness in disease management, particularly in controlling anthracnose [11]. In Brazil, numerous resistant cultivars have been developed for the management of major bean diseases, including anthracnose (*C. lindemuthianum*), angular leaf spot (*P. griseola*), and *Fusarium* wilt (*F. oxysporum* f. sp. *phaseoli*), among others [12].

Genome-wide association studies (GWAS) and QTL mapping are powerful approaches for elucidating genotype–phenotype associations underlying genetic resistance to fungal diseases in common bean. These analyses can be further strengthened by transcriptomic comparisons between tolerant and susceptible accessions, which facilitate the identification of potential resistance genes. This review highlights the major fungal diseases affecting common bean, with particular emphasis on those that cause the most severe damage to the crop. The most effective way to control these diseases is by utilizing genetic resistance, mainly through the cultivation of resistant varieties. The characterization of fungal pathogens is essential for developing cultivars with durable resistance. In this review, we report studies carried out on eleven fungal diseases affecting common beans and discuss their recent advances in molecular techniques. This review mainly focuses on genes and genomic regions associated with genetic resistance in common bean. It differs from previous reviews by emphasizing recent advances in molecular breeding, GWAS, QTL mapping, and the identification of novel candidate genes. The findings presented here provide valuable insights into the application of marker-assisted breeding aimed at improving resistance to fungal diseases in common beans.

The disease anthracnose (ANT) (*Colletotrichum lindemuthianum* (Sacc. & Magnus) Lams.–Scrib) causes symptoms on common bean like brown spots on the aerial part of the plant, frequently appearing on the lower leaf veins, stems, petioles, and pods. ANT may cause 100% yield loss when infected seeds are sown and there are favorable conditions for disease. The pathogen is present in the tropics and in temperate areas where temperatures range from 13 °C to 26 °C (optimum of 21 °C) and where high relative humidity (above 91%) and frequent rainfall occur, and this depends on whether the planting is done with infected and untreated seeds [13] (Figure 1).

The phytopathogen of ANT has several physiological races [14] and this facilitates the breakdown of resistance of cultivars and hinders their development. Fortunately, there are known sources of resistance alleles for this disease [15].

The prefix *Co* (derived from *Colletotrichum*) is a conventional notation used to identify genes conferring resistance to anthracnose in common bean. Beans from the Mesoamerican gene pool contain the resistance genes *Co*-2, *Co*-3 (*Co*-3^2^, *Co*-3^3^, *Co*-3^4^, and *Co*-3^5^ alleles), *Co*-4 (*Co*-4^2^ and *Co*-4^3^ alleles), *Co*-5 (*Co*-5^2^ allele), *Co*-6, *Co*-11, *Co*-16, *Co*-17, *Co*-u, and *Co*-v, which are mapped to chromosomes Pv02, Pv03, Pv04, Pv07, Pv08, and Pv11, respectively [16]. *Co loci* in the Andean gene pool are *Co*-1 (*Co*-1^2^, *Co*-1^3^, *Co*-1^4^, and *Co*-1^5^ alleles), *Co*-12, *Co*-13, *Co*-14, *Co*-15, *Co*-x, *Co*-w, *Co*-y, *Co*-z, *Co*-Pa, *Co*-AC, and *Co*Pv01CDRK, distributed across chromosomes Pv01, Pv03, Pv04, and Pv11. At the *Co*-1HY resistance *locus* [17], four genes were identified; among them, three serine/threonine protein kinases displayed elevated expression in resistant genotypes. According to the findings of Marcon et al. [18], the symbol *Co-Bf* was proposed to denote a new anthracnose resistance gene discovered in the Brazilian Andean common bean cultivar Beija Flor (Table 1).

Quantitative resistance *loci* (QRL) have also been mapped; López et al. [19] reported the identification of five QRLs spanning three chromosomes; Oblessuc et al. [20] found two major QRL (ANT02.1^UC^ and ANT07.1^UC^). Zuiderveen et al. [21] identified 14 QRLs with major effects on chromosomes Pv01, Pv02, and Pv04, and two additional QRLs with minor effects on Pv10 and Pv11. Wu et al. [22] mapped nine QRLs on Pv01, Pv02, Pv04, Pv05, Pv06, Pv10, and Pv11, whereas Perseguini et al. [23] identified 17 additional QRLs spanning Pv01–Pv08 and Pv11. Fritsche-Neto et al. [24] reported a QRL on Pv02 explaining 25% of the observed resistance.

Almeida et al. [25] analyzed a carioca diversity panel of 125 accessions genotyped with the BARCBean6K_3 BeadChip, alongside a segregating AND-277 × IAC Milênio population genotyped via GBS (genotyped by sequencing). Fourteen SNPs (single-nucleotide polymorphisms) were significantly associated with anthracnose resistance by GWAS (genome-wide association studies). The SNPs ss715642306 and ss715649427 at the beginning of chromosome Pv04 were identified, as well as 16 genes related to plant immunity. In the AND-277 cultivar, the Co-1^4^ *locus* was associated with resistance to ANT race 81, with three additional *loci* on Pv03, Pv10, and Pv11 also contributing to resistance.

Resistance in the CDRK (California Dark Red Kidney) cultivar is conferred by a single dominant *locus*, designated *Co*Pv01CDRK. Linkage analysis and GWAS mapped distinct *loci* on Pv04, Pv05, Pv10, and Pv11 controlling resistance to different isolates of race 65 [26]. A Yellow Bean Collection of 255 African genotypes was evaluated for resistance to *C. lindemuthianum* races 5, 19, 39, 51, 81, 183, 1050, and 1105 [27]. The Andean *locus Co*-1 on Pv01 was associated with resistance to races 81, 1050, and resistance to race 39 was mapped to Pv02. The Pv04 genomic region, with *loci* designated as *Co*-3, *Co*-15, *Co*-16, *Co*-y, and *Co*-z, was linked to resistance to races 5, 19, 51, and on Pv05 and Pv07, novel genomic regions associated with resistance to race 39 were identified.

According to Oblessuc et al. [28], the activation of innate immunity was studied in the SEL1308 bean genotype harboring the *Co-4^2^ locus*. Transcript levels of *PRb1a* and *PR1b* were upregulated at 72 and 96 h post-inoculation across all tissues involved in the incompatible interaction.

According to Lovatto et al. [29], the expression of plant defense genes was evaluated in the CDRK cultivar inoculated with race 73 of *C. lindemuthianum* by quantitative real-time PCR. The *Phvul.001G246300* gene showed the highest expression, encoding an abscisic acid (ABA) receptor with pyrabactin resistance, the PYL (*PYR1*-like) protein, which is central to the crosstalk between ABA and jasmonic acid (JA) signaling.

Chang et al. [30] identified 38 candidate genes related to disease resistance, comprising 10 *R* genes, 11 protein kinase genes, and 10 transcription factor genes. *Phvul.001G075600*, *Phvul.001G076200*, *Phvul.003G030000*, and *Phvul.010G073300* encode *R* proteins with kinase domains. The *WRKY* transcription factor gene *Phvul.008G251300* participates in the MAPK signaling pathway, while the *bZIP* gene *Phvul.003G291800* is involved in hormone signal transduction.

## 2. Angular Leaf Spot

Angular leaf spot (ALS) (*Pseudocercospora griseola* (Sacc) Crous & U. Brown) causes yield losses in common beans of up to 80% [31].

The disease occurs on cotyledonous and trifoliate leaves, stems, branches, petioles, and pods. On the leaves, the pathogen produces angular (circular) spots that are delimited by veins and are brown or reddish brown in color. On the abaxial leaf surface, grayish synemata (the fruiting of the pathogen) are observed, which can even lead to defoliation. Dark brown or reddish angular spots appear on the pod surface, without a defined center and with the presence of greyish-colored fruiting of the pathogen. In other parts of the plant (stem, branch, and petiole), elongated, dark brown lesions appear. Favorable factors for the occurrence of the disease include temperatures ranging from 16 °C to 28 °C (optimal at 24 °C), high relative humidity, and plants at the pod formation stages (R5 to R8) [15] (Figure 2).

The phytopathogen *P. griseola* also has different pathotypes [32], which can break cultivar resistance and hinder the development of new resistant genotypes.

ALS resistance in common bean is quantitative [33,34], yet five resistance *loci* have been identified by the BIC (Bean Improvement Cooperative) Genetic Committee: three dominant, independent *Phg loci* (*Phg*-1, *Phg*-2, *Phg*-3) and two major QTLs (*Phg*-4, *Phg*-5) [33,35].

The tight linkage (0.0 cM) between the *Phg*-1 *locus* and *Co*-1^4^ on Pv01 highlights their close genetic association in the Andean cultivar AND-277 [36]. Markers CV542014450 and TGA1.1570 flank the *Co*-1^4^/*Phg*-1 *loci* at 0.7 cM and 1.3 cM, respectively. AND 277 has been used in several studies investigating genetic control of resistance to various diseases [31].

The *Phg*-2 *locus* was mapped to chromosome Pv08 in the Mexico 54 and BAT cultivars [37]. The ALS resistance gene in BAT 332 was designated *Phg*-2^2^. The *Phg*-3 *locus* was mapped to Pv04 in the Ouro Negro cultivar [38]. A tight linkage (0.0 cM) was observed between the *Phg-3* and *Co-3^4^* alleles on chromosome Pv04 [38]. Two other major-effect QTLs have been reported: *Phg*-4 (ALS4.1^UC;GS^) on Pv04 and *Phg*-5 (ALS10^UC;GS^) on Pv10, both identified in the G5686 and CAL 143 cultivars [39] (Table 2).

Bean accessions evaluated in three PGS (plant growth stages) and environments exhibited varying levels of resistance, as the same cultivar considered resistant in one PGS showed susceptibility in another [31]. De Almeida et al. [31] assessed a BC_2_F_3_ population (AND-277 × IAC-Milênio, AM population) alongside a Carioca Diversity Panel (CDP). Both populations were evaluated for ALS resistance at the V2 and V3 PGSs (under controlled conditions) and at the R8 PGS (under field conditions). Multiple interval mapping detected seven significant QTLs, while GWAS indicated fourteen SNPs. *Phg*-1, *Phg*-2, *Phg*-4, and *Phg*-5 were detected in the CDP. The AND 277 cultivar carried both *Phg*-1 and *Phg*-5. At the beginning of chromosome Pv11, a novel QTL, designated ALS11.1^AM^, was identified. Within QTN2.1, a cluster of 21 NL (NB-LRR) genes was reported, and another cluster of five NL genes was located in QTN10.1. Additional NL genes found in other QTLs underscore their likely functional role as key candidates for ALS resistance.

The *Phg*-5 *locus* [40,41,42] has been the most extensively characterized, with its candidate genes thoroughly studied. The ALS10.1^UC^ core region contains 323 genes. GO (gene ontology) analysis revealed the presence of genes involved in signal perception and transduction, programmed cell death, and defense responses. These findings suggest that ALS10.1 may play a role in the early stages of the immune response to ALS. TIR-NB-ARC proteins are concentrated on Pv10, and most of them (51.1%) are encoded by genes in the ALS10.1^UC^ *locus*, highlighting the importance of these R genes in common bean response to *P. griseola*.

Lovatto et al. [29] assessed the expression of ten candidate genes adjacent to the *PhgPv01CDRK locus* in the CDRK cultivar following inoculation with race 63–39 of *Pseudocercospora griseola*. They found that the *Phvul.001G246300* gene exhibited the highest expression levels, as it was found for the *CoPv01CDRK/PhgPv01CDRK locus*. The gene encodes the ABA receptor PYL5, a protein that mediates interactions between ABA and JA signaling pathways.

## 3. Powdery Mildew

*Erysiphe polygoni* DC (*Oidium* sp.) most frequently occurs as the causal agent of powdery mildew (PM) in common bean [15]. However, other studies have also reported *Erysiphe diffusa* (Cooke & Peck) U. Braun & S. Takam as being associated with PM [43]. The pathogen affects the leaves, stems, and pods of the crop. On the upper leaf surface, the fungus causes small, dark spots followed by a white, powdery growth turning brown or purple and then yellowish, followed by defoliation. On the stems and pods, the fungus produces white powdery lesions, resulting in smaller pod size and malformation [15]. Temperatures from 20 °C to 25 °C and fluctuations between low and high humidity levels before flowering favor the occurrence of the disease [15] (Figure 3).

Two dominant resistance genes have been mapped in the common bean genome: *Pm*1, which confers complete resistance and is located at the end of chromosome Pv11, and *Pm*2, which confers intermediate resistance and is located on Pv04 [44]. *Pm*1 is epistatic over *Pm*2. These regions contain genes encoding proteins with NBS-LRR (nucleotide binding site—leucine-rich repeat) domains [45] (Table 3). Campa and Ferreira [46] identified a dominant gene physically mapped between 84,188 and 218,664 bp on chromosome Pv04 and proposed a candidate resistance gene, *Phvul.004G001500*, which encodes an EF (elongation factor). Oxidative burst and callose deposition have been associated with EFs and may play a role in mediating resistance to powdery mildew (PM).

Two QTLs, *PWM*2^AS^ and *PWM*11^AS^, were mapped on chromosomes Pv02 and Pv11, accounting for 7% and 66% of the phenotypic variation, respectively. The QTL on Pv11 was in the same genomic region as the ALS QTL, indicating a pleiotropic region. The peaks of the ALS11^AS^ and PWM11^AS^ QTLs on Pv11 coincide with the location of the BAR5054 marker. For PM, PWM11^AS^ accounted for 66% of the phenotypic variation [47].

Resistance to powdery mildew (PM) segregated at a single *locus* with codominant alleles, and the resistance *locus* was mapped to the proximal region of linkage group Pv04 [48]. This genomic region is rich in genes encoding typical plant *R* genes [49]. It also contains genes associated with complete resistance in Porrillo Sintetico [46] and partial resistance in Cornell 49242 [44].

SNPs were used to detect PM disease resistance in a panel of 211 bean genotypes [50]. Nine candidate genes were mapped on Pv04, Pv10, and Pv11. On Pv04, a gene cluster was found containing CC-NBS-LRR (coiled-coil nucleotide-binding site–leucine-rich repeat) and TIR-NBS-LRR (Toll/interleukin-1 receptor nucleotide-binding site–leucine-rich repeat)–type resistance genes. Two resistance genes, *Phavu_010G1320001g* and *Phavu_010G136800g*, were also identified on Pv10. Two genes on Pv11, *Phavu_011G167800g* and *Phavu_011G169300g*, reported as candidates for PM resistance, encode LRR (leucine-rich repeat) receptor-like serine/threonine protein kinases, suggesting a key functional role in pathogen defense.

## 4. Gray Mold

Gray mold is caused by *Botrytis cinerea* and causes considerable economic losses in common bean during harvest, storage and transportation [51]. On leaves, the pathogen produces dark, watery lesions with concentric rings and yellow margins, and with favorable conditions, it sporulates within the lesions, giving them a gray mold appearance. On stems and petioles, the pathogen causes longitudinal brown streaks. After the pathogen colonizes the flower, the pods become infected, developing watery, grayish-brown lesions, and under high humidity conditions, gray mold becomes visible on pod lesions [15]. Humid and cold periods (temperatures around 20 °C) favor the development of the disease. The disease may reach a critical level in dense crops and during the flowering and pod development stages [15] (Figure 4).

Gray mold resistance in beans involves JA and/or ET (ethylene) signaling pathways, which play key roles in activating the plant’s defense responses [52]. Other studies have indicated that ABA regulates plant defense against necrotrophs [53]. Nowogórska et al. [52] showed that BcNEP1 and BcNEP2 toxins from *B. cinerea* associate with plasma membranes and the nuclear envelope, and can enter the nuclei of plant cells, where they likely play a role in triggering host responses. These toxins cause high accumulation of hydrogen peroxide, evident in chloroplasts. Van Baarlen et al. [54] reported that during *Botrytis* sp. colonization, ROS (reactive oxygen species) production in the cell wall is often accompanied by phenolic compound synthesis, contributing to the plant’s defense responses.

*Botrytis* virulence exhibits a polygenic inheritance [55,56,57]. It was reported that significant SNPs linked to the disease were not within genes nor introns but were mostly in intergenic regions [57]. In that study, *Botrytis* isolates were evaluated across a range of eudicot hosts.

Fungal small RNAs can suppress host immunity by triggering gene silencing [58]. Mengiste et al. [59], in a study on *A. thaliana*, observed that *B. cinerea* infection activates the *BOS1* gene. *BOS1* mutants, which are more susceptible to *B. cinerea*, also exhibit increased sensitivity to oxidative stress induced by abiotic factors. Moreover, *BOS1* expression was absent in plants carrying the *coi1* mutation, which confers JA insensitivity and increased susceptibility to *Botrytis*. The protein encoded by *BOS*1 (R2R3MYB transcription factor), together with JA, may participate in a signaling cascade leading to ROI (reactive oxygen intermediates) activation.

Expression of the *SIAIM*1 gene identified in tomato plants (*Solanum lycopersicum*) is regulated by ABA [60]. Plants with reduced expression of this gene proved to have greater susceptibility to both *B. cinerea* infection and salt stress. AbuQamar et al. [61] also stated that the *EXLA*2 gene, which encodes an expansin involved in cell wall alteration, plays an important role in *B. cinerea* infection. The lack of *EXLA*2 gene expression or its deregulation may be noteworthy to amplify resistance to *B. cinerea*.

## 5. *Fusarium* wilt

*Fusarium oxysporum* f. sp. *phaseoli (Fop)* causes yield losses in common bean of up to 100% [62]. The soil-borne fungus Fop can survive in soil organic matter, crop residues, and the roots of some non-host plants [63].

In common bean, the Fop infection process begins at the root tip [64], followed by obstruction of xylem vessels, which leads to temporary and persistent wilting of the leaves, vascular discoloration, progressive yellowing from lower to upper leaves, early defoliation, dwarfism, and premature plant death [65,66]. The disease may affect the leaves and branches of only one side of the plant, due to localized colonization of vessels (Figure 5).

Favorable factors for the occurrence of the disease include temperatures ranging from 20 °C to 28 °C, water stress, sandy and acidic soils, nematode (*Meloidogyne* sp.) infestation in cultivation areas, and crops in the early vegetative to flowering stages [15].

The genetic inheritance and potential sources of Fop resistance in common bean have been explored in various studies. Fall et al. [66] identified a major QTL in recombinant inbred lines infested with Fop race 4, accounting for 63.5% of phenotypic variation, highlighting its potential utility in breeding for Fusarium resistance.

Cavalheiro [67] applied DArT (Diversity Arrays Technology) in the F_2_ generation of a cross between BRS Notável (resistant) and BRS Supremo (susceptible) in a field naturally infested with Fop, identifying a stable QTL on Pv07 that explained 44.8% of the phenotypic variation. Using the same technology on the F_2_ population of a cross between BRS FP403 (resistant) × BRS Horizonte (susceptible), Torres et al. [68] identified QTLs on chromosomes Pv01, Pv02, Pv03, and Pv04 that explained 5.8% to 40.5% of the variation. A TaqMan assay using probes that targeted two SNPs associated with resistance on chromosome Pv02 achieved a selection efficiency of 92%, highlighting its potential for MAS (marker assisted selection).

SNPs linked to *Fusarium* wilt resistance in common bean accessions have been identified using GWAS [69,70,71]. In a panel of 133 Portuguese accessions, associative mapping for Fop race 6 revealed nine SNPs linked to resistance on Pv04, Pv05, Pv07, and Pv08 [69]. These authors and others have reported that Fop resistance is polygenic, involving multiple genes with small effects [69,70,71,72].

Paulino et al. [70] detected SNPs associated with resistance to the Fop UFV01 isolate on Pv05 and Pv11 using 205 Mesoamerican bean varieties and measuring the area under the disease progress curve (AUDPC). SNP ss715648096 on Pv11 was additionally linked to disease severity. Another SNP on Pv03 also proved to have a significant association with the disease severity rating. Regarding the IAC18001 isolate, significant SNPs were identified on Pv03, Pv04, Pv05, and Pv07, as well as on Pv01, Pv05, and Pv10.

In a panel of 157 common bean accessions (including 21 Fop-resistant lines), Chiwina et al. [71] identified SNPs linked to Fop resistance: 16 on Pv04, Pv05, Pv07, Pv08, and Pv09 for race 1; 23 on Pv03, Pv04, Pv05, Pv07, Pv09, Pv10, and Pv11 for race 4; and seven on Pv04 and Pv09 conferring resistance to both races.

Elevated expression of genes involved in flavonoid biosynthesis has been described, as well as genes responsive to SA (salicylic acid), JA, and ethylene hormones [73]. Ethylene-responsive genes were associated with the final stages of Fop infection [73]. PvMES1 was associated with increasing resistance to Fop through the SA pathway [74]. PvMES1 upregulated several genes in the phenylpropanoid route (lignins, flavonoids, and cinnamaldehyde) during Fop infection (Table 4).

PvTGA03 and PvTGA07 were found to participate in SA, taking on opposite roles in response to *Fusarium* wilt [75]. These genes exhibit high expression in common bean roots. Fop infection expanded H_2_O_2_ accumulation, which is associated with tissue damage. Intensive ROS accumulation promotes pathogen colonization and xylem cell damage in susceptible roots [76].

## 6. Dry Root Rot

This disease is caused by the soilborne phytopathogenic fungus *Fusarium solani* f. sp. *phaseoli*. The pathogen causes reddish-brown longitudinal streaks without defined margins on the hypocotyl and primary roots, which progress into older parts of the roots and necrotize the entire root system as well as necrotic longitudinal fissures in the roots and collar region. Plant leaves turn yellow, and the plant may die. Interactions with the fungus *R. solani* have been reported, causing damping-off, wilting, and plant death [15]. Severely attacked plants can be easily uprooted from the soil [15] (Figure 6).

Favorable factors for the occurrence of the disease include excessive soil moisture, soil compaction, and mild temperatures (18 °C to 22 °C). Water deficit and acidic soils can stress plants and facilitate the occurrence of *F. solani* f. sp. *phaseoli* (FSP) in plants [15]. The disease is most critical in compacted soils and during the early vegetative growth stages (V1 to V3) [15].

In Mesoamerican recombinant inbred lines from Puebla 152 × Zorro, a QTL on Pv05 was identified for resistance to FSP [77]. Hagerty et al. [78] reported a QTL for FSP resistance on Pv03 in a snap bean RIL (recombinant inbred line) population. Román-Avilés and Kelly [79] detected nine QTLs explaining 5–53% of the phenotypic variation in two populations derived from crosses between a Mesoamerican black bean and an Andean kidney bean, and between the same black bean and an Andean cranberry bean. A GWAS performed with field data from an ADP (Andean diversity panel) and 3525 SNPs identified a 3.3 Mbp region on Pv04 associated with FSP resistance [80]. Sixteen genomic regions were identified for the ADP and seven for the MDP (Mesoamerican diversity panel) by Zitnick-Anderson et al. [81]. Candidate genes for the ADP included those encoding a glucan synthase-like enzyme, senescence-associated proteins, and NAC domain proteins. For the MDP, the main candidate genes were the ethylene response factor 1 and the MAC/Perforin domain-containing gene.

## 7. *Pythium* Root Rot

The phytopathogen *Pythium ultimum* (*P. ultimum* var. *ultimum*) is responsible for root rot (PRR) in common bean [82]. The species *P. ultimum*, *P. irregulare*, and *P. paroecandrum* are active in environments with temperatures below 20 °C, while *P. myriotylum* and *P. aphanidermatum* occur in warmer regions, between 20 °C and 35 °C. PRR causes seed rot (before germination), damping-off, root rot, and consequently, leaf yellowing and plant wilting [83].

In young plants, the fungus causes watery lesions on the collar region and even in the roots, which then become necrotic, destroying the roots and killing the plant. Surviving plants exhibit underdevelopment, chlorosis, wilting, and reduced yield [15]. Pods close to the moist soil and infected may develop soft rot and a white mycelial growth of the fungus. In the early vegetative stages of the crop (stages V0 to V3), the incidence of the disease is high [15] (Figure 7).

Regarding genetic resistance to *Pythium* in common bean, a major gene (*Py-1*) has been proposed to be involved in the genetic control of resistance to seed decay and pre-emergence damping-off. This gene is located on Pv07 and is closely linked to the *P* gene, which is involved in the control of seed coat color and is associated with flavonoid compounds [84]. Other studies have associated increased resistance to *P. ultimum* with colored-seed genotypes [85,86].

Genomic regions conferring resistance to *Pythium* spp. were mapped on Pv01, Pv02, Pv04, Pv05, and Pv09 using the isolate MS61, an ADP, and both 260K GBS-based and 6K BeadChip SNP markers [82]. Candidate genes included *Phvul.002G119700*, located 16.97 kb from marker S02_25507837 (25.50 Mb) and encoding a subtilase family protein, and *Phvul.002G278400*, located near marker ss715645959 (44.79 Mb) and encoding a DEFL (defensin-like) protein involved in plant defense responses.

## 8. Southern Blight

The soilborne phytopathogen *Sclerotium rolfsii* Sacc. [teleomorph *Athelia rolfsii* (Curzi) C. C. Tu & Kimbr.] is responsible for the disease known as stem rot or southern blight, which occurs in common beans grown in tropical and subtropical regions around the world [87]. In warmer regions and during rainy periods followed by dry weather, the incidence of the disease can range from 30% to 100% [88]. Common symptoms of southern blight in the plant include water-soaked gray to brown lesions on the stem above the soil line, which progress to stem decay and destruction of the main root. Because of this rot, the aerial parts of the plant show yellowing of the lower leaves, which gradually progresses upward.

Severely affected plants have constricted root collars, leading to wilting, leaf drop, and plant death. Under high soil moisture conditions, white fungal mycelium with characteristic dark brown, small spherical sclerotia appears on the plant root collar [89]. Seedling tipping symptoms may also occur in early crop stages [15]. Conditions that favor the development of the disease in the vegetative stages of the crop are high temperatures from 27 °C to 30 °C, high humidity, and low soil pH [15] (Figure 8).

The most effective strategy to limit damage caused by collar rot is the use of resistant cultivars. *Sclerotinia sclerotiorum* and *Sclerotium rolfsii* exhibit similar modes of myceliogenic germination of sclerotia [90], comparable growth habits [91], and the release of oxalic acid during infection. SCAR (sequence-characterized amplified region) markers developed for white mold resistance were used to evaluate 30 improved landraces of Guilan dwarf bean [87]. Among them, Pach-02 and Pach-08 were reported as susceptible and resistant to *Sclerotium rolfsii*, respectively, while 19 genotypes were tolerant. The authors reported that although all SCAR markers successfully identified the resistant genotype, only the SMe1Em5 marker was able to distinguish susceptible from tolerant genotypes.

## 9. White Mold

Yield losses of up to 85% in common bean can be caused by white mold in some regions of the world [92]. In Brazil, depending on the location, the damage caused by this disease can range from 85% to 100% [91]. The soilborne fungus *Sclerotinia sclerotiorum* (Lib.) of Bary is responsible for white mold in common bean.

The pathogen infects leaves, petioles, branches, stems, and pods of the crop, causing watery lesions (waterlogging) that spread quickly, evolving into soft rot, where plant tissues turn yellow and then brown. Under high humidity, rotting is abundant, with the presence of white, cottony mycelium from the fungus plus sclerotia (a resistance structure). This structure can be produced on plant parts and in the soil and survives in the absence of the crop for 6 to 8 years. These structures, as well as planting infected seeds, can be sources of disease inoculum [15].

Due to soft rot in the stem, branches, and petioles, the foliage of the aerial part becomes wilted, and the branches die [15] (Figure 9). Pods close to the soil and infected by the phytopathogen produce discolored and smaller seeds.

The environmental factors favorable for the occurrence of the disease are temperatures between 15 °C and 25 °C, high relative humidity, and frequent rainfall. Crops that are too densely planted or have close row spacing cause shading and poor aeration within the plant canopy, and this creates conditions conducive to the development of the fungus [15]. Under these conditions, the fungus produces apothecia in the soil. This is a fungal structure containing many sexual spores that are ejected into the air, carried by the wind, and deposited on plant tissues, causing disease.

The incidence of the disease is frequent in winter, in fields with center-pivot irrigation, or during rainy periods, especially when the crop is at the flowering stage (R6) and pod development stage (R7) [15]. The phytopathogen *S. sclerotiorum* survives on crop residues and is polyphagous, as it can infect more than 300 plant species.

Small- to moderate-effect QTLs have been reported on all linkage groups except Pv10 [93]. Several testing methods identified 27 QTLs for disease resistance and 36 for disease avoidance across different biparental populations [94]. A prior study of 14 recombinant inbred biparental populations derived from 12 dry bean and 2 snap bean parents detected 37 QTLs associated with resistance. A meta-analysis of these populations subsequently identified 17 distinct QTLs [95]. GWAS has refined QTL mapping, closing minor gaps and revealing new QTL [96]. Arkwazee et al. [97] performed GWAS on snap beans and identified 34 SNPs associated with white mold resistance—the seedling straw test revealed 11 SNPs, and 23 SNPs were from field data. The most common candidate genes were PPR (pentatricopeptide repeat)-encoding genes and LRR-encoding genes. PPRs are recognized as disease-resistance gene analogs [98], whereas LRRs play key roles in effector-triggered and PAMP (pathogen/microbe-associated molecular pattern)-triggered plant immunity [98].

## 10. Ashy Stem Blight/Charcoal Rot

Charcoal rot is caused by the soilborne phytopathogenic fungus *Macrophomina phaseolina* (Tassi) Goidanich. The most common symptoms of the disease in bean seedlings and adult plants are pre- and post-emergence damping-off and the formation of dark, well-defined lesions on the stem. The stem lesions cause yellowing and wilting of the foliage, which may be more pronounced on just one side of the plant. In more developed plants, the fungus causes stunting, leaf chlorosis, premature defoliation, and more depressed and darker stem lesions, with abundant black pycnidia and microsclerotia of the fungus. Stem injury can cause the plant to fall, while pods in contact with the soil can become infected with the fungus, causing seed contamination. Planting contaminated seeds can result in seedling damping-off [15]. Environmental factors that favor the occurrence of the disease are high temperatures (above 27 °C) and low soil moisture, with plants in the vegetative to early reproductive stages [15] (Figure 10). The pathogen can also infect soybean, sorghum, maize, cotton, and several other crops.

Ashy stem blight (ASB) resistance can be inherited qualitatively or quantitatively, depending on the genetic background of the resistant host, the screening method used, and the environmental conditions. Two complementary dominant genes (*Mp*-1 and *Mp*-2) were identified as conferring resistance in the BAT 477/A 70 F_2_ population evaluated in a growth chamber [99]. Mayek-Pérez et al. [100] identified two dominant genes exhibiting double recessive epistasis, along with nine QTLs from BAT 477, as contributors to field resistance against *Macrophomina phaseolina*. Furthermore, nine QTLs on Pv03, Pv05, Pv06, Pv08, Pv09, and Pv10 were associated with *M. phaseolina* resistance in both field and controlled environments in a BAT 477 × UI-114 RIL population [101]. Miklas et al. [102] identified QTL on Pv04, Pv06, Pv07, and Pv08 through field evaluation for resistance to ASB in the Dourado × XAN 176 RIL population, descending from resistance from the XAN 176 black bean. Moreover, Viteri and Linares [103] reported two recessive genes conferring resistance to *M. phaseolina* in the PC 50 × Othello cross, and one recessive gene in the Badillo × PR1144-5 cross, based on greenhouse evaluations. These genes were derived from the Andean genotypes PC 50 and Badillo. The authors also reported a dominant gene associated with resistance in the A 195 × PC 50 population (Table 5).

Recently, 107 F_6:7_ RILs derived from a cross between the susceptible cultivar Verano and the partially resistant breeding line PRA154 were genotyped using 109,040 SNPs [104]. A major QTL on Pv07, accounting for 40% of the phenotypic variance, was identified. Candidate genes for ASB resistance included *Methylcrotonyl-CoA carboxylase alpha chain* and *MCCA* (mitochondrial 3-methylcrotonyl-CoA carboxylase 1). Recombinant inbred lines (126 F_6:7_) derived from the cross between BAT 477 and NY6020-4 and 72,017 SNPs were used in conducting GWAS for ASB resistance. A novel QTL region was found on Pv03, and a candidate gene was detected encoding a drought-sensitive, WD repeat-containing protein 76 (Phvul.003G175900) [105].

## 11. Rust

The causal agent of rust in common bean is the obligate biotrophic fungus *Uromyces appendiculatus* F. Strauss. The pathogen infects leaves and pods, and occasionally stems and branches. Symptoms first appear on the abaxial leaf surface as small, light-colored spots, which develop into mature uredinia that are brownish-red and rupture the epidermis [106]. The disease is more severe in humid tropical and subtropical regions and during the vegetative to flowering stages of the crop [15]. Isolates of *U. appendiculatus* can exhibit extremely high virulence diversity, with more than 90 races of the fungus described globally [107] (Figure 11).

Several studies have been conducted to enhance the genetic resistance of common bean to rust and include identification and mapping of resistance genes, screening of germplasm for resistant genotypes, and development of molecular markers to support breeding programs.

Eleven dominant resistance genes have been identified: six genes (*Ur*-3, *Ur*-3+, *Ur*-5, *Ur*-7, *Ur*-11, and *Ur*-14) are from the Mesoamerican pool and five genes (*Ur*-4, *Ur*-6, *Ur*-9, *Ur*-12, and *Ur*-13) are from the Andean pool [108] (Table 6).

The *Ur-3* gene, which confers resistance to 55 of 94 rust races and is present in the Mesoamerican white-seeded common bean cultivar Aurora, was fine-mapped to chromosome Pv11, with the SS68 KASP (Kompetitive Allele Specific PCR) marker found to be tightly linked to it [107,108]. The *Ur-6* gene, present in the Golden Gate Wax cultivar, was fine-mapped by GWAS to a 251.5 Kb interval on chromosome Pv07, where five candidate genes involved in plant–pathogen defense were identified [109]. The *Ur*-11 gene, from the Mesoamerican PI 181996 cultivar, was mapped by GWAS to a 400 Kb interval on chromosome Pv11, where a gene encoding an LRR-containing protein was identified as the most likely candidate [110]. *Ur-11* confers wide-ranging resistance to most races of *Uromyces appendiculatus*, being ineffective only against the Honduran race 22. Therefore, it can be used in combination with other resistance genes as a cost-effective strategy for control of the disease. Valentini et al. [111] also mapped *Ur-11* to a 9.01 Kb genomic region on chromosome Pv11, flanked by markers SS322 and SS375. Marker SS322 exhibited 97.5% accuracy in detecting the presence of *Ur-11* in common bean plants.

Novel sources of partial and incomplete hypersensitive resistance to rust were identified by GWAS in a collection of Portuguese germplasm [112]. Partial resistance was associated with chromosome Pv08, while incomplete hypersensitive resistance was linked to chromosomes Pv06, Pv07, and Pv08, suggesting oligogenic inheritance for both traits.

Table 7 summarizes the major fungal diseases affecting common beans, the plant resistance mechanisms against these diseases, the main symptomatology, and the risk factors that favor their occurrence.

## 12. Breeding Stories Based on Molecular Breeding

MAS has shown great potential for gene introgression, particularly when DNA polymorphisms are maximized in wide crosses between gene pools. Molecular markers are already available for several major diseases, including angular leaf spot (ALS) [35] and anthracnose (ANT) [16].

MAS has been effectively employed to introgress the anthracnose resistance genes *Co-5* and *Co-4^2^*. The SCAR markers SAB3 and SAS13, which are linked to *Co-5* and *Co-4^2^*, respectively, in the resistant donor genotype G2333, enabled efficient identification and selection of resistant progeny during backcrossing with susceptible commercial cultivars [113].

Resistant donor lines such as G2333 and other genotypes carrying *Co loci* (e.g., *Co*-4, *Co*-5) have been used to transfer anthracnose resistance into susceptible Andean climbing and carioca-type backgrounds through marker-assisted backcrossing and pyramiding. This approach generated resistant lines that maintained the target grain type preferred by the market [114].

QTLs explaining substantial variation in white mold resistance were identified on chromosomes Pv7 and Pv8, and corresponding molecular markers were established. These markers were employed to introgress the QTLs into elite, yet susceptible, pinto bean backgrounds. Breeders successfully combined physiological resistance with architectural (avoidance) traits, thereby improving field performance under disease pressure [115].

Key rust resistance genes of historical importance, such as *Ur-3* and *Ur-11*, have been fine-mapped, providing valuable insights for marker-assisted breeding [108,110,111]. Linked markers were developed and validated, enabling breeders to efficiently track and pyramid these genes into new cultivars. Fine-mapping efforts also improved marker reliability and precision for MAS applications.

The *Phg*-2 *locus*, conferring ALS resistance, was fine-mapped to a small genomic region, and breeder-friendly markers were validated through field trials. These markers are currently used to introgress *Phg*-2 into susceptible germplasm and to pyramid multiple ALS resistance genes [115].

A recent breeding program employed a marker-assisted parallel backcross (MAPBC) scheme to pyramid *Phg*-2 (ALS resistance) with two major QTLs for common bacterial blight (CBB) into a preferred Ethiopian cultivar [116]. The resulting lines exhibited enhanced resistance to CBB and ALS under field conditions.

Field evaluations comparing lines carrying single versus pyramided ALS resistance genes demonstrated that gene stacking increased both resistance level and stability [117]. The crosses involved AND277, Mexico 54, G5686, and two susceptible cultivars, K132 and Kanyebwa. To confirm the presence of five ALS resistance genes, SCAR and SSR (Simple Sequence Repeat) markers were applied to tag the pyramided genes in F_2_ progenies. Four-gene combinations in four-parent crosses conferred stronger resistance to ALS isolate 61:63 than combinations containing only two or three genes.

Additionally, the SCAR marker OAY15.200 was identified as the closest marker to *Ur*-6, at a genetic distance of 7.7 cM [118]. This marker offers an important tool for pyramiding rust resistance genes in common bean (Figure 12).

## 13. Conclusions and Future Perspectives

Molecular breeding of common bean cultivars faces the dual challenge of meeting the demands of a growing global population while adapting to the changing conditions imposed by climate change. In addition, fungal pathogens of the bean crop continue to evolve, often overcoming existing host resistance mechanisms, which underscores the need for continuous innovation in breeding strategies. It is nearly impossible to develop a single common bean cultivar that is resistant to all fungal pathogens. However, understanding the underlying genetic architecture can provide critical guidance for implementing MAS strategies.

Recent progress in common bean improvement has predominantly emphasized molecular marker technologies, such as Illumina single-nucleotide polymorphism (SNP) arrays, kompetitive allele-specific PCR (KASP) assays, genotyping-by-sequencing (GBS), and various high-throughput sequencing platforms. The availability of a complete common bean reference genome provides an invaluable resource for breeding programs, and databases such as Phytozome (https://phytozome-next.jgi.doe.gov/info/Pvulgaris_v2_1, accessed on 9 September 2025) offer targeted genomic information to support these efforts.

The integration of pan-genomes and high-quality reference sequences enables the identification of functional resistance alleles, the validation of candidate genes, and the potential for precise gene editing using CRISPR-Cas9 technology. Genomic selection combined with comprehensive marker panels allows breeders to pyramid multiple resistance *loci* within breeding populations, while speed-breeding approaches accelerate generation turnover, iteratively combining molecular validation with field selection. Collectively, these advances have the potential to address the challenges posed by climate change, pathogen evolution, and rising global demand, thereby enhancing the resilience and productivity of common bean cultivation.

## Figures and Tables

**Figure 1 ijms-26-10387-f001:**
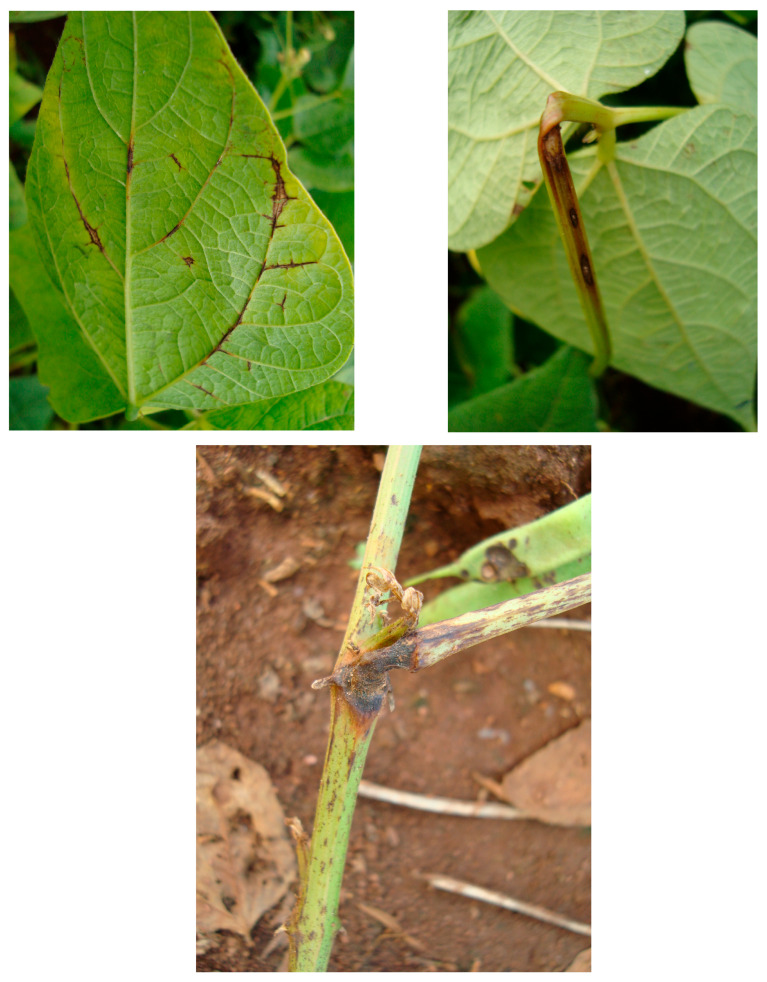
Anthracnose caused by *Colletotrichum lindemuthianum* in common bean [13].

**Figure 2 ijms-26-10387-f002:**
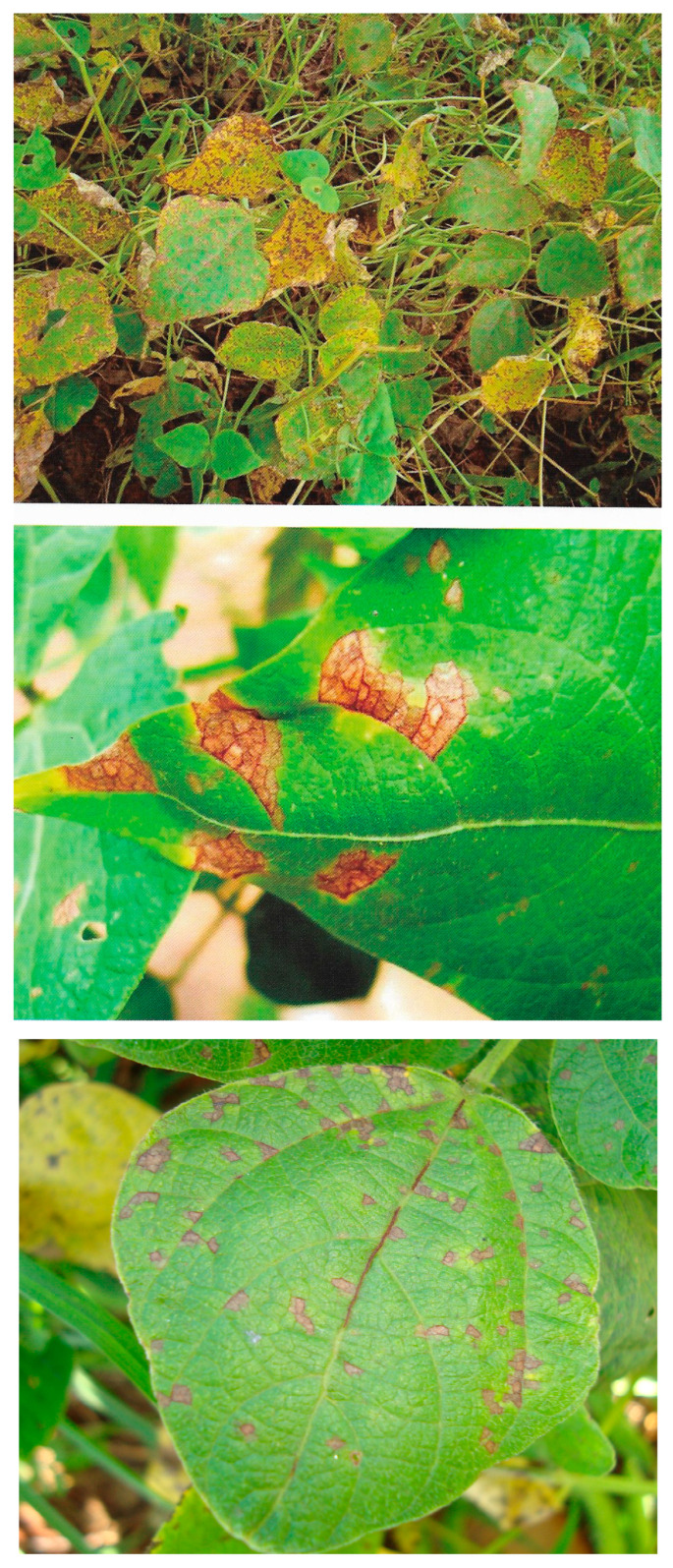
Angular leaf spot caused by *Pseudocercospora griseola* in common bean [13].

**Figure 3 ijms-26-10387-f003:**
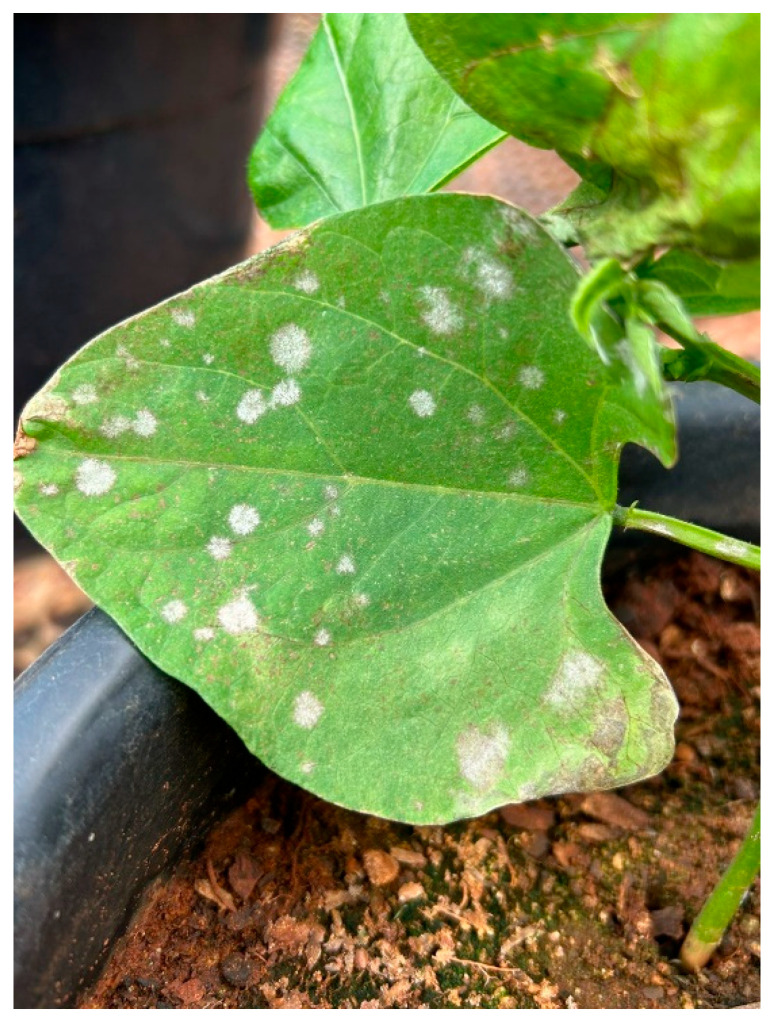
Powdery mildew caused by *Erysiphe poligony* in common bean [13].

**Figure 4 ijms-26-10387-f004:**
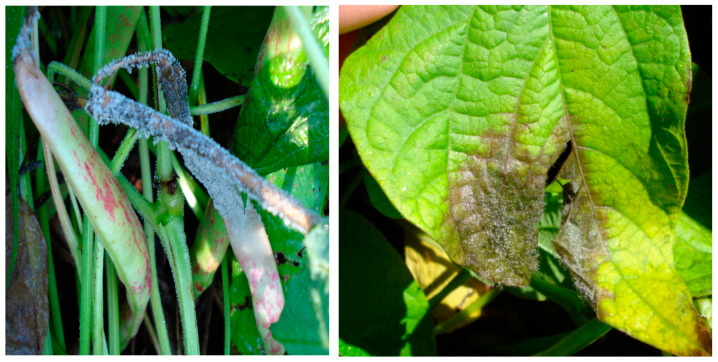
Grey mold caused by *Botrytis cinerea* in pods and common bean leaf [13].

**Figure 5 ijms-26-10387-f005:**
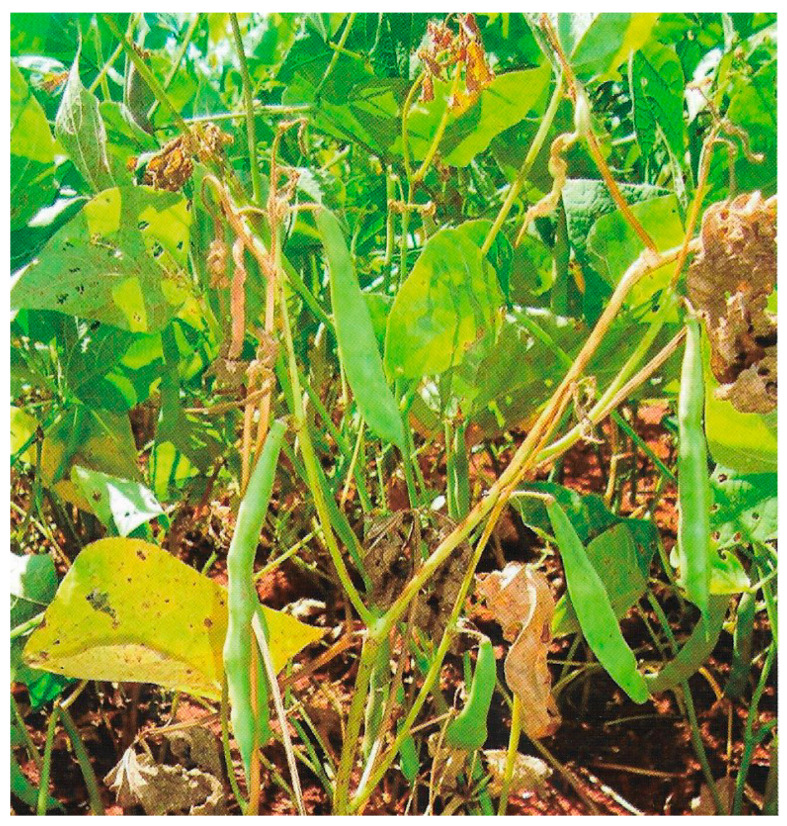
*Fusarium* wilt caused by *Fusarium oxysporum* f. sp. *phaseoli* in common bean [13].

**Figure 6 ijms-26-10387-f006:**
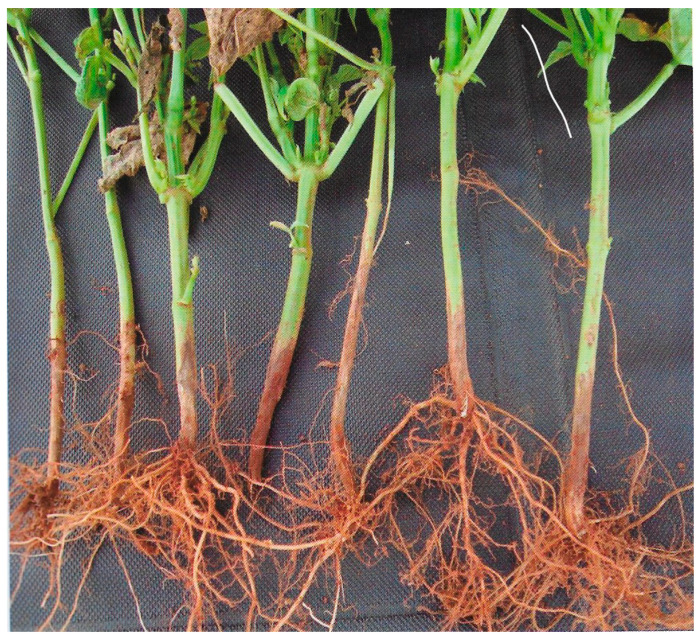
Dry Root Rot caused by *Fusarium solani* f. sp. *phaseoli* in common bean [13].

**Figure 7 ijms-26-10387-f007:**
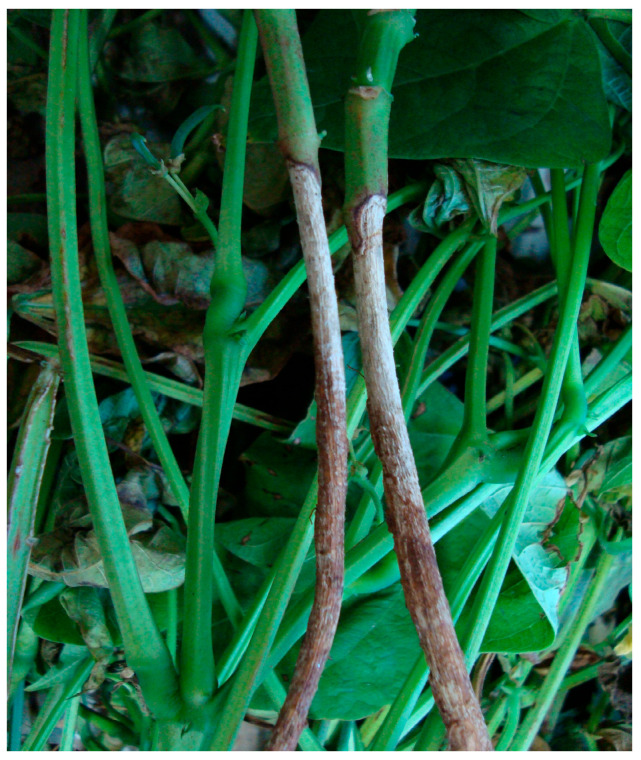
*Pythium* occurring in the basal region of the common bean stem [13].

**Figure 8 ijms-26-10387-f008:**
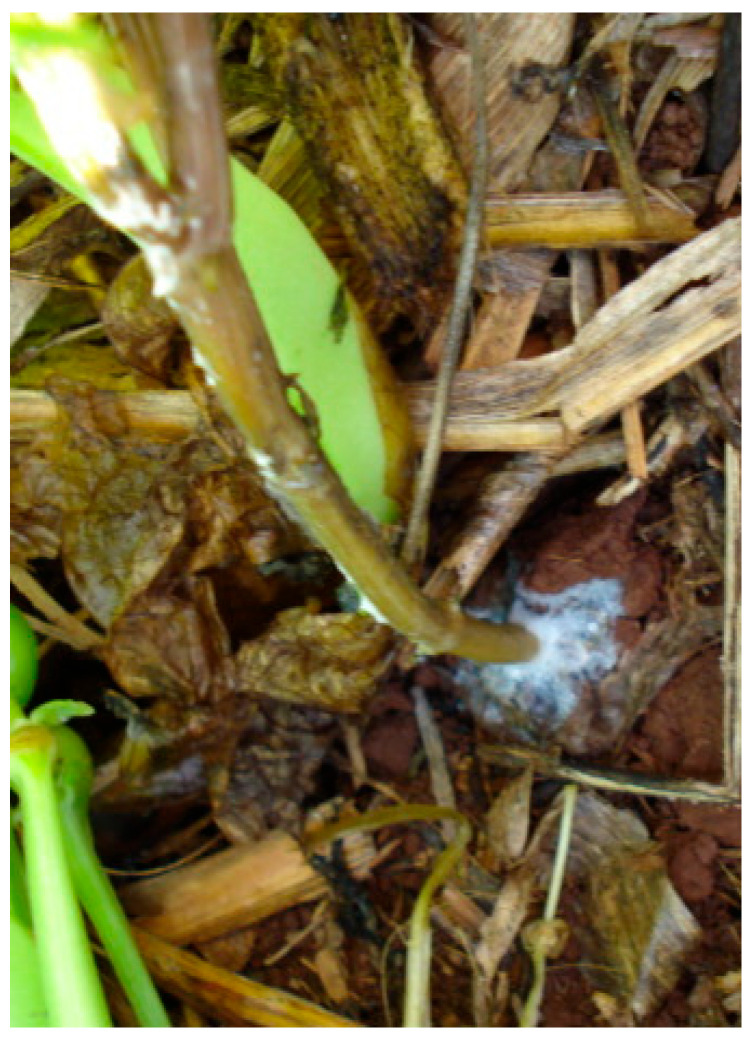
Southern blight caused by *Sclerotium rolfsii* in the collar region of common bean [13].

**Figure 9 ijms-26-10387-f009:**
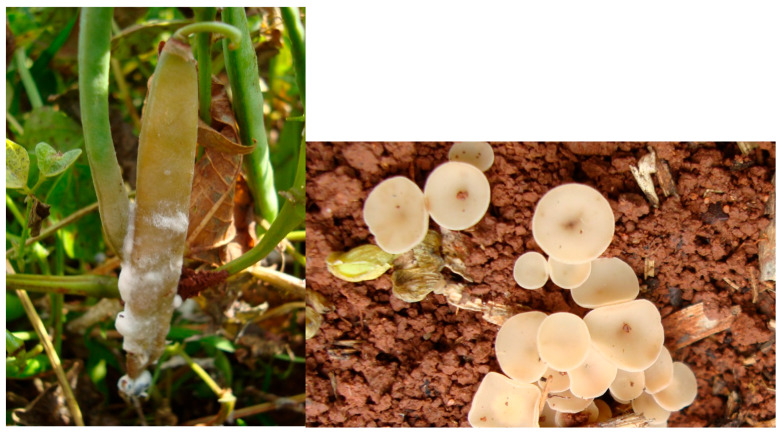
White mold (*Sclerotinia sclerotiorum*) in close-to-soil bean pod (**left**) and pathogen apothecia in soil (**right**) [13].

**Figure 10 ijms-26-10387-f010:**
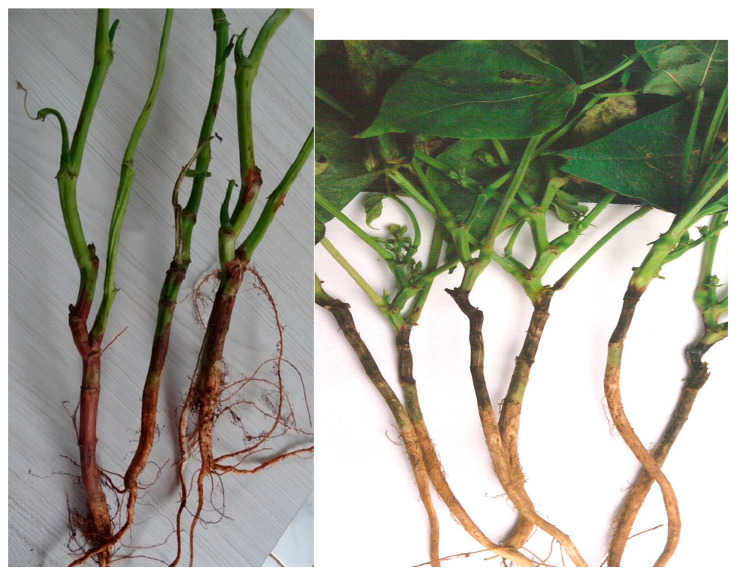
Ashy stem blight caused by *Macrophomina phaseolina* in common bean [13].

**Figure 11 ijms-26-10387-f011:**
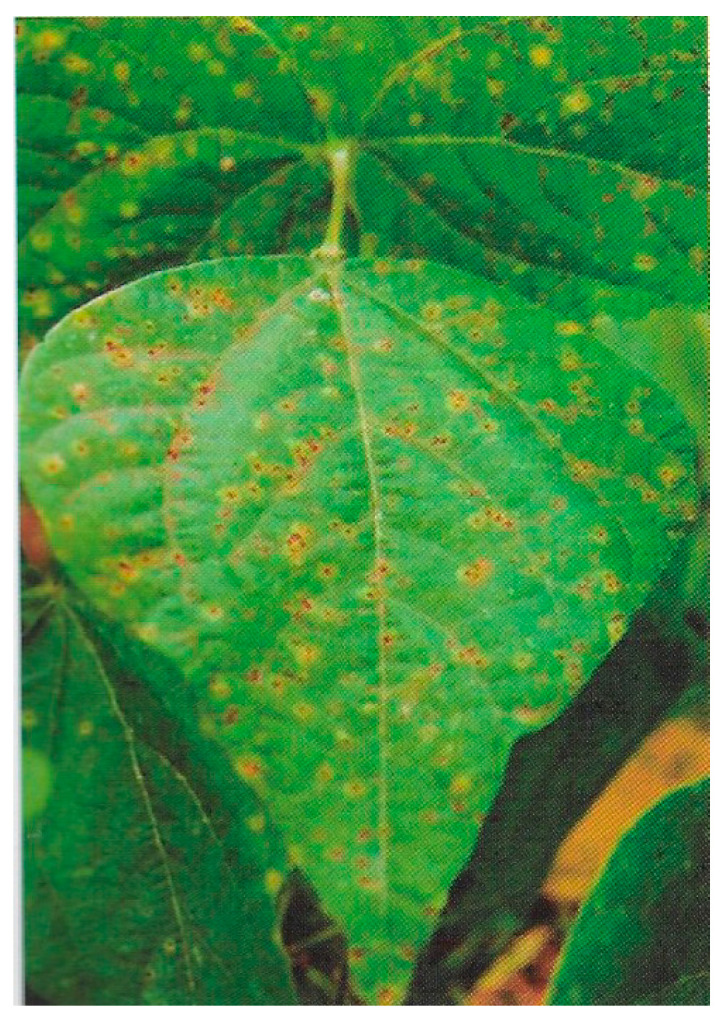
Rust caused by *Uromyces appendiculatus* in common bean [13].

**Figure 12 ijms-26-10387-f012:**
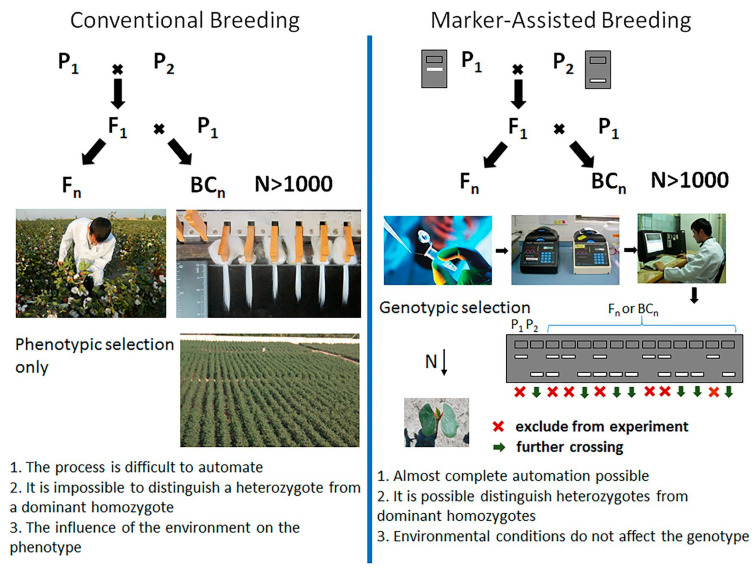
A schematic comparison between conventional breeding and marker-assisted breeding [119].

**Table 1 ijms-26-10387-t001:** Main *Co* (*Colletotrichum*) genes for anthracnose resistance.

Gene	Pv	Breeding Implications
*Co*-1	Pv01	It was identified in the Andean cultivar Michigan Dark Red Kidney (MDRK).
*Co*-2	Pv11	It was identified in the Andean cultivar Michigan Dark Red Kidney (MDRK).
*Co*-3	Pv04	It was identified in the Mesoamerican cultivar Mexico 222. *Co*-3 alleles were identified in other sources, such as G2333.
*Co*-4	Pv08	It was originally identified in the To (Tu) Andean cultivar.
*Co*-5	Pv07	It was described in the Mesoamerican cultivar Mexico 222.
*Co*-6	Pv11	It was identified in the Mesoamerican cultivar AB 136.
*Co*-12	Pv04	It was identified in the Mesoamerican cultivar Michelite.
*Co*-13	Pv03	It was reported in the Brazilian Ouro Negro Mesoamerican cultivar.
*Co*-14	Pv01	It was reported in the Andean cultivar AND 277.
*Co*-15	Pv04	It was identified in the Andean cultivar AND 277.
*Co*-16	Pv04	It was described in the Mesoamerican cultivar Bat 93.
*Co*-17	Pv04	It was described in the Andean cultivar Crioulo 159.
*Co*-u	Pv02	It was described in the Mesoamerican cultivar Mexico 222.
*Co*-v	Pv07	It was reported in the Brazilian Andean cultivar Rudá.
*Co*-x	Pv01	It was mapped and associated with the Andean cultivar Jalo EEP558.
*Co*-y	Pv04	It was mapped and associated with the Andean cultivar Jalo EEP558.
*Co*-w	Pv01	It was mapped in the distal part of Pv01.
*Co*-z	Pv04	It was identified in the Andean Brazilian cultivar Zorro.
*Co*-Pa	Pv01	It was related in the Andean Brazilian cultivar Paloma.
*Co*-AC	Pv01	It was identified in the Brazilian Andean Paloma cultivar. This gene is mapped on the Pv01 chromosome, specifically in a region close to other resistance genes, such as *Co*-1, positioned at the end of the chromosome.
*Co*Pv01CDRK	Pv01	It was related to the Andean cultivar AND 277. This gene is an allele of *locus Co*-14.
*Co*-1HY	Pv01	It was identified and cited in the Andean cultivar Hongyundou, which has strong resistance to the 81 race of *Colletotrichum lindemuthhianum*. The resistance is conferred by a dominant gene located at the *Co*-1 *locus*.
*Co*-Bf	Not mapped	It was identified and cited in the Andean Brazilian cultivar Beija Flor.
*Co*-F2533	Pv06	It was identified and described in the F2533 genotype.

**Table 2 ijms-26-10387-t002:** The five ALS resistance *loci* that have been recognized by the BIC (Bean Improvement Cooperative) Genetic Committee.

Gene/QTL	Pv	Breeding Implications
*Phg*-1	Pv01	It was identified in the Andean cultivar AND 277.
*Phg*-2	Pv08	It was described in the Mexico 54 and BAT cultivars.
*Phg*-3	Pv04	It was reported in the Ouro Negro cultivar.
*Phg*-4	Pv04	A QTL was identified and cited in the G5686 and CAL 143 cultivars.
*Phg*-5	Pv10	A QTL was identified in the G5686 and CAL 143 cultivars.

**Table 3 ijms-26-10387-t003:** Dominant genes conferring resistance to Powdery mildew.

Gene	Pv	Breeding Implications
*Pm*1	Pv11	It was identified in Cornell 49242, a Mesoamerican gene pool cultivar.
*Pm*2	Pv04	It was reported in Cornell 49242, a Mesoamerican gene pool cultivar.
Phvul.004G001500	Pv04	The resistant cultivar Porrillo Sintético has differential expression of the gene.
Phavu_010G1320001g	Pv10	It was found in a diversity panel.
Phavu_010G136800g	Pv10	It was described in a diversity panel.
Phavu_011G167800g	Pv11	It was identified in a diversity panel.
Phavu_011G169300g	Pv11	It was found in a diversity panel.

**Table 4 ijms-26-10387-t004:** Genes related to *Fusarium* wilt resistance.

Gene	Pv	Breeding Implications
*PvMES1*	Not mapped	It was found in the resistant CAAS260205 genotype.
*PvTGA03*	Pv03	It was described in the resistant CAAS260205 genotype.
*PvTGA07*	Pv09	It was reported in the resistant CAAS260205 genotype.

**Table 5 ijms-26-10387-t005:** Genes for resistance to *Macrophomina phaseolina*.

Gene	Pv	Breeding Implications
*Mp-1*	Not mapped	It was identified in resistant accession BAT-477.
*Mp-2*	Not mapped	It was reported in resistant accession BAT-477.

**Table 6 ijms-26-10387-t006:** Dominant resistance genes identified to *Uromyces. appendiculatus* resistance in common bean.

Gene	Pv	Breeding Implications
*Ur*-3	Pv11	It was identified in the resistant parent Aurora (Mesoamerican).
*Ur*-4	Pv06	The cultivar Early Gallatin is the differential with Ur-4 (Andean).
*Ur*-5	Pv04	It was described in Mexico 309 (Mesoamerican).
*Ur*-6	Pv7	The Olathe cultivar and Golden Gate Wax cultivar are reported to carry *Ur*-6 (Andean).
*Ur*-7	Pv11	It was identified in Great Northern 1140 (GN1140, Mesoamerican).
*Ur*-9	Pv01	The PC-50 or Pompadour Checa 50 is the Andean landrace that carries *Ur*-9.
*Ur*-11	Pv11	The accession PI 181996 (a Guatemalan black bean) is a classic source (Mesoamerican).
*Ur*-12	Pv07	The PC-50 or Pompadour Checa 50 is the Andean landrace that carries *Ur*-12.
*Ur*-13	Pv3	The Kranskop (resistant, Andean) cultivar was used as the donor parent in marker development.
*Ur*-14	Pv04	It was reported in the cultivar Ouro Negro (Mesoamerican).

**Table 7 ijms-26-10387-t007:** Summary table of the diseases described reflecting the main mechanisms of plant resistance, main symptoms and risk factors.

Disease	Mechanisms of Plant Resistance	Symptomatology	Risk Factors
Anthracnose	Resistance genes (*Co*) and QTLs of minor effects (Pv01, Pv02, Pv03, Pv04, Pv05, Pv06, Pv07, Pv08, Pv10, and Pv11).	Brown spots on the aerial part of the plant, frequently appearing on the lower leaf veins, stems, petioles, and pods.	*Colletotrichum lindemuthianum* has many physiological races. Temperatures around 18 °C and high humidity favor the disease.
Angular leaf spot (ALS)	Resistance genes and QTLs (*Phg*). ALS in common bean is quantitative (Pv01, Pv04, Pv08, Pv10, and Pv11).	On the leaves, the pathogen produces angular spots that are delimited by veins and are brown or reddish brown in color.	The *P. griseola* phytopathogen has different pathotypes. Temperatures around 16 °C to 28 °C (optimum 24 °C) and high humidity favor the disease.
Powdery mildew	Seven dominant resistance genes and QTLs on Pv02, Pv04, Pv10, and Pv11.	On the upper leaf surface, the phytopathogen causes small, dark spots followed by a white, powdery growth turning brown or purple and then yellowish, followed by defoliation. On the stems and pods, the fungus produces white powdery lesions, resulting in smaller pod size and malformation	*Erysiphe poligony* presents huge genetic variability. Temperatures around 20–25 °C and low to high humidity changes favor the occurrence of the disease.
Gray mold	*Botrytis* virulence has a quantitative nature.	On leaves, the pathogen produces dark, watery lesions with concentric rings and yellow margins, and with favorable conditions, it sporulates within the lesions, giving them a gray mold appearance.	*Botrytis cinerea* occurs on bean plants when there are humid periods and cold temperatures (around 20 °C).
*Fusarium* wilt (Fop)	Fop resistance is polygenic, involving multiple genes with small effects. Three resistance genes were found and QTLs (Pv01, Pv02, Pv03, Pv04, Pv05, Pv07, Pv08, Pv09, Pv10, and Pv11).	The infection begins at the root tip and then obstructs and discolors the vascular system, leading to temporary and persistent wilting of the leaves, progressive yellowing from the lower to the upper leaves, stunting, premature leaf drop, and early plant death.	*Fusarium oxysporum* f. sp. *phaseoli* exhibits races and high variability. Factors favorable for the occurrence of the disease in common bean are temperatures ranging from 20 °C to 28 °C, water stress, sandy and acidic soils, as well as the presence of nematodes.
Dry Root Rot	QTLs on Pv03, Pv04 and Pv05.	The pathogen causes reddish-brown longitudinal streaks without defined margins on the hypocotyl and primary roots, which progress into older parts of the roots and necrotize the entire root system as well as necrotic longitudinal fissures in the roots and collar region.	*Fusarium solani* f. sp. *phaseoli* is favored by excess soil moisture, soil compaction, and mild temperatures in the range of 18 °C to 22 °C.
*Pythium* root rot (PRR)	There is a major gene (*Py*-1) on Pv07 and QTLs identified on Pv01, Pv02, Pv04, Pv05, and Pv09.	PRR causes seed rot (before germination) and root rot, damping-off, and, consequently, leaf yellowing and plant wilting.	*Pythium ultimum* is most active at temperatures below 20 °C. *P. aphanidermatum* develops more from 20 °C to 35 °C.
Southern blight	SMe1Em5 marker could differentiate susceptible from tolerant genotypes.	The symptoms of southern blight are water-soaked gray to brown lesions on the stem above the soil line, which, over time, cause stem rot and destruction of the main root.	The factors that favor the development of the disease (*Sclerotium rolfsii*) are crops in the vegetative stage, high temperatures of 27 °C to 30 °C, high humidity, and low soil pH.
White mold	Small- to moderate-effect QTLs have been reported in all linkage groups except Pv10.	The pathogen infects leaves, petioles, branches, stems, and pods of the crop, causing watery lesions (waterlogging) that spread quickly, evolving into soft rot, where plant tissues turn yellow and then brown.	The environmental conditions favorable for the development of *Sclerotinia sclerotiorum* in common bean are temperatures between 15 °C and 25 °C, high relative humidity, and frequent rainfall.
Charcoal rot	Resistance can be inherited qualitatively or quantitatively. Two complementary dominant genes (*Mp*-1 and *Mp*-2) were identified. QTL on Pv03, Pv04, Pv05, Pv06, Pv07, Pv08, Pv09, and Pv10 were also identified.	In seedlings and adult plants, the disease causes pre- and post-emergence damping-off and the formation of dark, well-defined lesions on the stems. The stem lesions result in yellowing and wilting of the plant’s foliage.	The factors favorable for the development of *Macrophomina phaseolina* in common bean are high temperatures (above 27 °C), low soil moisture, and plants in the vegetative to early reproductive stages.
Rust	Eleven dominant resistance genes and QTLs have been identified on Pv01, Pv03, Pv04, Pv06, Pv07, Pv08 and Pv11.	Initial symptoms appear on the abaxial leaf surface as small, light-colored spots, which develop into mature, brownish-red uredinia that rupture the epidermis.	Isolates of *Uromyces appendiculatus* exhibit extremely high virulence diversity, with more than 90 races of the fungus described. The disease is more severe in humid tropical and subtropical regions and occurs when the plant is in the vegetative to flowering stages.

## Data Availability

No new data were created or analyzed in this study. Data sharing is not applicable to this article.

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
