# Peer review of "Molecular Breeding for Fungal Resistance in Common Bean"

_ijms, 2025, doi:10.3390/ijms262110387_

Round 1

Reviewer 1 Report

Comments and Suggestions for Authors

Author Response

1. The review is a summary of a huge literature but does not necessarily point out the novelty of this synthesis. What is the difference between this review and previous reviews (e.g. focus on recent developments in molecular breeding, synthesis of GWAS and QTL mapping, or novel candidate genes)?
R: We have accepted the reviewer´s comment and added information in lines 108 to 112.
2. The abstract is informative and it appears more like an introduction. It ought to highlight the major conclusions of the review, and provide a more precise take-home, on how molecular tools are transforming fungal resistance breeding in beans.
R: We have accepted the reviewer’s comment and added information on lines 21 to 32.
3. Although the introduction covers the significance of beans, it can be furthered by making explicit why fungus resistance is an urgent breeding issue (e.g. loss of yield due to climate change, restriction of fungicides, development of pathogens).
R: We have accepted the reviewer´s suggestion and added more information in lines 55-72 and lines 84-104.
4. Both parts are very detailed, though the story is frequently a long list of genes and QTLs. To make it easier to read, I would include summary tables (e.g., “Major resistance loci identified in each disease, their chromosomal location, and breeding implications).
R: We have accepted the reviewer’s comment and added six tables with the main dominant genes for anthracnose (Table 1), angular leaf spot (Table 2), powdery mildew (Table 3), fusarium wilt (Table 4), charcoal rot (Table 5) and rust (Table 6). The QTLs information was gathered in Table 7.
5. The manuscript has outlined GWAS and QTL mapping individually per disease. Perhaps it would be handy to discuss the convergent nature of various methods (GWAS, transcriptomics, candidate gene validation) to find resistance loci.
R: We included the information in the introduction section (Lines 98-104).
6. The article could use more discussion concerning the ways that breeders can apply this knowledge. An example is to point out certain markers that have been validated in other populations and are currently being applied in MAS (marker-assisted selection). The majority of the photos are photos of diseases. Although it is helpful, the paper could use at least one schematic figure that can be used to face the molecular breeding pipeline (QTL/GWAS to candidate genes to MAS/pyramiding to resistant cultivars).
R: We agreed with reviewer´s suggestion and added information on a marker development that could be used for MAS (Lines 668-707) and a schematic figure (Figure 12).
7. The conclusion section is short. Expand with specific suggestions (e.g. pyramiding resistance loci, combining genome editing, pan-genomics, genomic selection or speed breeding in common bean). There are a few too long and hard to read sentences. As an illustration, the description of the life cycles of pathogens or QTL mapping outcomes might be made shorter and easier. To an adequate English editing, I would recommend. Some abbreviations are defined without explanation (e.g. ROI, ET in initial
chapters). Defining all abbreviations where they are first used would be wise.
R: We have expanded conclusion section and corrected the English writing of the sentences. We also checked the abbreviations throughout the manuscript. All the abbreviations were defined when they are first used.
8. Consider including a table with all major fungal pathogens of beans including their distribution worldwide, impact on yield and whether resistance genes/QTL have been identified. This would aid readers to position themselves easily with the landscape.
R: We added information in Table 7 according to Reviewer´s #2 suggestion.
9. It contains a number of small typographical and formatting errors (e.g., h ps://doi.org in place of https://doi.org). Also, do check the manuscript closely. Some disease (e.g., anthracnose, ALS) are talked about in great detail whereas other diseases (e.g., southern blight, gray mold) are rather short. Think about how much you can cover or make it clear as to why you are covering certain ones in greater detail. It would be more effective and useful to researchers and breeders to include a brief sub-section, called Breeding Success Stories (managed to release resistant cultivars based on molecular breeding) to increase the impact and applicability of the review.
R: We have corrected the small typographical and formatting errors in the reference section. We included a ‘Breeding stories based on molecular breeding’ as a topic of the article (Lines 668 to 707).

Reviewer 2 Report

Comments and Suggestions for Authors

The review by Luciana Lasry Benchimol-Reis and co-authors is devoted to the molecular aspects of bean resistance to fungal diseases. Fungi are among the most formidable enemies of plants in general and crops in particular, so this review is undoubtedly of great value to specialists in plant protection in the relevant field. The authors pay great attention to describing the genetic aspects that determine bean resistance to various fungal diseases.

I have a few minor comments, which in no way detract from the value of this review, but will greatly simplify the process of understanding it for potential readers.

1. Like any review, this manuscript contains a lot of detailed information, which obscures the integrity and main message of the authors. This work is devoted to fungal diseases, and it would be very useful to see a summary table of the diseases described, reflecting the main mechanisms of plant resistance, characteristics of infection, risk factors, etc.

2. I have questions about figures 1 and 7. They appear to be stretched horizontally, which makes the overview look messy. I may be mistaken, but I would ask the authors to pay attention to these figures.

3. Some fungi, despite causing enormous damage to plants, look very aesthetic. Perhaps such masterful photos can be found to replace figures 3 (Erysiphe poligony) and 6 (Sclerotinia sclerotiorum).

4. I would like to see specific values that would allow to assess the damage caused by fungal diseases. For example, in a certain currency, as a percentage of lost harvest, as well as a comparison with viral and bacterial diseases.

Author Response

The review by Luciana Lasry Benchimol-Reis and co-authors is devoted to the molecular aspects of bean resistance to fungal diseases. Fungi are among the most formidable enemies of plants in general and crops in particular, so this review is undoubtedly of great value to specialists in plant protection in the relevant field. The authors pay great attention to describing the genetic aspects that determine bean resistance to various fungal diseases.
I have a few minor comments, which in no way detract from the value of this review, but will greatly simplify the process of understanding it for potential readers.

1. Like any review, this manuscript contains a lot of detailed information, which obscures the integrity and main message of the authors. This work is devoted to fungal diseases, and it would be very useful to see a summary table of the diseases described, reflecting the main mechanisms of plant resistance, characteristics of infection, risk factors, etc
R: We organized Table 7 with the suggested information.
2. I have questions about figures 1 and 7. They appear to be stretched horizontally, which makes the overview look messy. I may be mistaken, but I would ask the authors to pay attention to these figures.
R: Figure 1 and Figure 7 were reformulated. Other figures were added so that each disease has its own figure.
3. Some fungi, despite causing enormous damage to plants, look very aesthetic. Perhaps such masterful photos can be found to replace figures 3 (Erysiphe poligony) and 6 (Sclerotinia sclerotiorum).
R: We altered Figures 3 and 6 according to the reviewer’s suggestion.
4. I would like to see specific values that would allow to assess the damage caused by fungal diseases. For example, in a certain currency, as a percentage of lost harvest, as well as a comparison with viral and bacterial diseases.
R: We have included the information requested in lines 57-72.

Round 2

Reviewer 1 Report

Comments and Suggestions for Authors

Although the authors have made commendable efforts to revise and improve the manuscript in accordance with the provided suggestions, the similarity index remains relatively high (33%). It is recommended that the authors further reduce the level of textual similarity to ensure the manuscript meets the journal’s publication standards before proceeding to the next stage

Author Response

Campinas, October 21st, 2025.

Editor

Dr. Andrés J. Cortés

Dear Dr.,

It is with great pleasure that we present our manuscript “Molecular breeding for fungal resistance in common bean” for your consideration. We have improved the English and diminished the similarity index. We hope that now it can be accepted in International Journal of Molecular Sciences special issue ‘Plant Breeding and Genetics: New Findings and Perspectives’.

Sincerely yours,

Luciana Lasry Benchimol-Reis

Agronomic Institute (IAC)

Plant Genetic Resources Centre

Zip Code 13075-630

Campinas, SP, Brazil

e-mail: luciana.reis@sp.gov.br

Phone:(+5519) 971452318
